# Current Status and Future Perspectives of Checkpoint Inhibitor Immunotherapy for Prostate Cancer: A Comprehensive Review

**DOI:** 10.3390/ijms21155484

**Published:** 2020-07-31

**Authors:** Tae Jin Kim, Kyo Chul Koo

**Affiliations:** 1Department of Urology, C.H.A. Bundang Medical Center, University College of Medicine, Seongnam 13496, Korea; tjkim81@cha.ac.kr; 2Department of Urology, Gangnam Severance Hospital, Yonsei University College of Medicine, Seoul 06229, Korea

**Keywords:** biomarkers, clinical trials, immune checkpoint inhibitor, immunotherapy, prostatic neoplasm

## Abstract

The clinical spectrum of prostate cancer (PCa) varies from castration-naive to metastatic castration-resistant disease. Despite the administration of androgen synthesis inhibitors and chemotherapy regimens for castration-resistant prostate cancer, the treatment options for this entity are limited. The utilization of the immune system against cancer cells shows potential as a therapeutic modality for various solid tumors and hematologic malignancies. With technological advances over the last decade, immunotherapy has become an integral treatment modality for advanced solid tumors. The feasibility of immunotherapy has shown promise for patients with PCa, and with advances in molecular diagnostic platforms and our understanding of immune mechanisms, immunotherapy is reemerging as a potential treatment modality for PCa. Various combinations of individualized immunotherapy and immune checkpoint blockers with androgen receptor-targeted therapies and conventional cytotoxic agents show promise. This article will review the current status of immunotherapy, including new discoveries and precision approaches to PCa, and discuss future directions in the continuously evolving landscape of immunotherapy.

## 1. Introduction

Prostate cancer (PCa) is the most commonly diagnosed malignancy among men and the second most common cause of cancer-associated death in industrialized nations [1]. The standard primary treatments for localized PCa are radical prostatectomy or radiation therapy (RT) with or without androgen-deprivation therapy (ADT), while the primary treatment modality for PCa patients in the advanced setting is ADT. Despite initial therapeutic responses with ADT, the majority of patients are destined to progress to metastatic castration-resistant PCa (mCRPC) [2]. Therefore, novel therapeutic approaches with durable response rates for advanced disease are warranted.

Current agents approved by the US Food and Drug Administration (FDA) that have shown efficacy in terms of overall survival (OS) in patients with mCRPC include docetaxel, cabazitaxel, radium-223, sipuleucel-T, and androgen receptor axis-targeted agents including abiraterone and enzalutamide [3,4,5,6,7,8]. Since the approval of sipuleucel-T in 2010 [4], there have been significant advancements and innovations in the field of immunotherapy. However, with the exception of sipuleucel-T, no other immunotherapeutic treatment has been approved for the treatment of mCRPC, owing to limited response rates and modest clinical efficacy. Nevertheless, with the introduction of next-generation genomic diagnostic platforms and advancements in our understanding of the molecular pathophysiology, we are now observing a considerable shift in the paradigm of immunotherapy for the treatment of PCa.

This review will highlight the re-emergence of immunotherapeutic approaches in the treatment of PCa, mainly focusing on immune checkpoint inhibitors (ICIs), which have shown the potential to revolutionize the treatment of both localized and advanced PCa.

## 2. Pathophysiology of Prostate Cancer

The pathogenesis of PCa is a slow, ongoing progression that involves small developing cancers that gradually evolve into different clonal entities with various clinical outcomes [9,10,11]. Previous studies have shown that chronic inflammation is frequent in the prostates of elderly men and that this change is associated with an increased risk of PCa development [12,13]. The underlying mechanisms of chronic prostate inflammation and its clinical relevance to the development of PCa are unclear. Nevertheless, clinical evidence shows that chronic inflammation is a potential risk factor for disease progression and poor clinical outcomes [14,15,16,17].

Cellular pathways that modulate the proliferation, migration, and survival of cancer cells are upregulated through cytokines and signaling molecules, including tumor necrosis factor-alpha (TNF-α), transforming growth factor-beta (TGF-β), C-C motif chemokine ligand 2 (CCL-2), and interleukins (IL-2, IL-6, IL-8, and IL-10) [17]. T cell activity and the associated autoimmune reaction induce epithelial and stromal cell proliferation. Infiltrating B cells are associated with increased antibody production, and recent studies have reported that specific subsets, namely regulatory B cells, may also play a significant role in tumor progression [18,19,20]. Moreover, increased B cell infiltrates have been observed in PCa [21]. Although T and B cells are the immune cell types that are predominantly observed in PCa, various ILs and inflammatory cell cytokines that dwell in the tumor stroma, such as granulocytes, macrophages, natural killer (NK) cells, myeloid-derived suppressor cells (MDSCs), and monocytes, additionally play an essential role in promoting the autocrine or paracrine proliferation of PCa cells [22,23,24,25].

Along with the aforementioned immune cells, androgen receptors (ARs) are a crucial factor in the tumorigenesis of PCa. The AR is a ligand-dependent transcription factor that is densely populated in thymic epithelial cells (TECs) and is essential for the development and function of male accessory sex organs [26,27]. ARs have a crucial role in the proliferation of PCa cells. The downregulation of gene expression required for regular TEC activity results in decreased thymocyte proliferation and an increase in apoptosis, which ultimately leads to thymic involution [27]. AR signaling directly affects the activity of circulating T cells by increasing the transcription of the protein tyrosine phosphatase non-receptor type 1 (PTPN 1). This downregulates the Janus kinase/signal transducers and activators of transcription (JAK/STAT) signaling pathway and subsequently suppress T helper type 1 (Th1) cell differentiation [28]. Moreover, androgen cessation can inversely affect the involution of the thymus by upregulating thymocyte proliferation and differentiation and can subsequently stimulate T cell infiltration [29,30]. Evidence from numerous studies has shown that PCa cells consist of a highly reactive stroma, which is featured by upregulated T cell infiltrates densely populated with regulatory T cells that have immunosuppressive tendencies [31,32].

As integral members of the innate immune system, myeloid cells can be stratified into two individual groups. The first group consists of mononuclear cells such as dendritic cells, macrophages, and monocytes, and the second consists of polymorphonuclear cells, which include basophils, eosinophils, mast cells, and neutrophils. Many studies have shown a tumor-promoting function of terminally differentiated myeloid-derived cells, as they can promote cancer cell proliferation and migration, tumor angiogenesis, and immune suppression [33,34,35,36,37]. In the PCa microenvironment, the cessation of androgen induces CCL2 upregulation and activation of the STAT 3 pathway, which initiates the recruitment of macrophages and induces polarization of M2 macrophages [38,39,40]. Activated M1 macrophages have the ability to promote a Th1 response and can exert an efficient antigen-presenting function with the capability of terminating tumors. However, by initiating the proliferation of epithelial and stromal cells, neovascularization, and suppression of the immune system, the anti-inflammatory characteristics of M2 macrophages stimulate the Th2 response and tissue remodeling, all of which creates favorable changes in the microenvironment for disease progression [41].

## 3. Immunotherapy Resistance of Prostate Cancer

Various clinical trials of PCa immunotherapy have been targeted at metastatic disease, with specific advances being made for novel therapies for CRPC. Patients with metastatic PCa have a dysfunctional and compromised immune system [42]. The immunogenicity of the cancer cells is the main determining factor in the immunotherapeutic approach to tumor eradication. Cancer cells are integrated into the self-immune system and, therefore, do not express foreign antigens. However, cancer cells exhibit tumor self-antigens, which can be immunogenic. Tumor neoantigens are the result of somatic mutations that are amassed in dividing cancer cells and are clinically correlated with the degree of mutation [43]. The load of tumor mutation has been associated with clinical outcomes of immunotherapy; however, tumor immunogenicity is influenced by numerous components that are regulated by the cancer cells [44,45,46,47,48,49].

Patients with metastatic PCa are known to have disrupted cellular immunity as well as a tumor microenvironment with increased immunosuppressive qualities. The compromised immune system of patients with PCa is characterized by a reduction in NK cell activity and renewal and inhibited expression of CD3 in NK and T cells, which may ultimately result in the reduction of T cell receptors and NK cell-activating receptors [50,51,52]. Moreover, patients with mCRPC also have a reduced number of total T cells [53] and an increased number of myeloid suppressor cells and regulatory T cells in the tumor microenvironment and circulation [54,55,56]. Another potential explanation for the resistance and tolerance to immunotherapy in PCa is the slow disease progression [57,58]. The low mutational tumor burden in patients with PCa may contribute to de novo immunotherapy resistance (Figure 1) [59]. This view, however, is still under debate, as a recent genomic analysis indicated that patients with PCa had a higher tumor mutation burden than patients with renal cell carcinoma [60].

Further research is warranted to elucidate the exact pathological mechanism underlying the resistance of patients with PCa to immunotherapy to pave the way for an effective immunotherapeutic regimen for metastatic PCa.

## 4. Immune Checkpoint Inhibitors for the Treatment of Prostate Cancer

The development of ICIs has been one of the most dynamic advances during the last decade. To date, sipuleucel-T, a cancer vaccine based on autologous antigen-presenting cell (APC) immunotherapy, is the only immunotherapeutic agent that is FDA-approved for the treatment of PCa [4]. ICIs, which are antibodies inhibiting the immune checkpoint receptors, cytotoxic T lymphocyte-associated protein 4 (CTLA-4), and programmed death 1/programed death ligand 1 (PD-1/PD-L1), have shown potential therapeutic benefits by producing antitumor effects and long-term survival benefits in a broad spectrum of malignancies (Figure 2) [61,62,63]. However, numerous clinical trials in patients with mCRPC utilizing CTLA-4 or PD-1/PD-L1 inhibitors have been unsatisfactory, with limited survival benefit when administered as a monotherapy in unselected patients [64,65,66,67,68]. The sole exception is the FDA-approved usage of pembrolizumab for all malignancies with mismatch repair (MMR) deficiency or a microsatellite instability (MSI)-high status [69,70]. To overcome the therapeutic limitation, combination regimens consisting of ICIs with other modalities, or in a specific subset of patients who would most likely benefit, have resulted in better treatment efficacies in patients with a specific subtype of mCRPC.

### 4.1. ICI Monotherapy

Ipilimumab, a monoclonal antibody blocking the immune properties of CTLA-4, was the first effective ICI to show potential efficacy for melanoma [71]. The therapeutic success of ipilimumab and its FDA-approval in 2011 triggered an interest in its use in PCa. An open-label phase I/II multicenter clinical trial (NCT00323882) investigated ipilimumab with or without RT in patients with mCRPC [72]. After stratifying the patients into two individual dose escalation arms, the study endpoints included adverse events (AE), prostate-specific antigen (PSA) decline, and tumor response. A select number of patients in both study groups exhibited antitumor activity as measured by PSA response. Eight (16%) patients in the combination arm (10 mg/kg) showed a PSA decline >50%, while one (2.0%) patient showed a complete response (CR) lasting 11 months. Six (12.0%) patients had stable disease which lasted for 2.8 to 6.1 months [72]. Encouraging clinical evidence of antitumor activity in previous studies incited further examination of ipilimumab in several phase III clinical trials. In the CA184-043 phase III trial (NCT00861614), patients with mCRPC who had at least one bone metastasis following taxane-based chemotherapy were allocated to either ipilimumab or placebo following palliative RT every three weeks [67]. Although the study barely missed the primary endpoint in terms of OS between the ipilimumab group and the placebo group (HR 0.85, CI 0.72–1.00; *p* = 0.053), progression-free survival (PFS) was significantly superior in the ipilimumab group. Moreover, subset analysis revealed that ipilimumab was the most therapeutically efficacious in the subgroup of patients who exhibited favorable prognostic characteristics, including normal serum alkaline phosphatase levels, normal hemoglobin concentration, and no visceral metastases. The OS outcomes in this subgroup of patients were superior with ipilimumab compared to placebo [67]. In a recent randomized, double-blind phase III trial (NCT01057810), ipilimumab monotherapy was compared to a placebo in patients with asymptomatic or minimally symptomatic chemotherapy-naïve mCRPC [65]. The median OS was 28.7 months in the ipilimumab group and 29.7 months in the placebo group (*p* = 0.367). Median PFS was 5.6 months in the ipilimumab group and 3.8 months in the placebo group (HR = 0.67; 95.87% CI 0.55–0.81). This trial concluded that ipilimumab failed to prolong OS in patients with mCRPC. However, after observing a longer median PFS in the ipilimumab group compared to the placebo group (5.6 months vs. 3.8 months) and a decline in serum PSA values, the investigators concluded that these findings suggest the presence of antitumor activity in a specific subgroup of patients [65]. Indeed, future studies are required to determine how to measure such antitumor activity, preferably with biomarkers, to identify the subgroup of patients who would most benefit from ipilimumab.

Several studies focused on PD-1/PD-L1 inhibition in patients with metastatic PCa. In a phase Ib trial of nivolumab (NCT00730639), 296 patients with non-small cell lung cancer, advanced melanoma, renal cancer carcinoma, or CRPC received nivolumab every two weeks [68]. Favorable objective response rates (ORRs) were shown in patients with melanoma, non-small cell lung cancer, and renal cell carcinoma. However, there were no responses observed in 17 patients with CRPC. PCa samples collected from the patients yielded negative results for PD-L1 expression [68]. Martin et al. focused on the deficiency of PD-L1 expression in PCa and emphasized another challenge in the ongoing battle of PCa treatment [73]. Contrary to the phase I nivolumab trial, the non-randomized phase Ib KEYNOTE-028 (NCT02054806) clinical trial only included patients with PD-L1 expression in ≥1% of tumor or stroma cells [66]. Patients received pembrolizumab 10mg/kg every two weeks for up to two years, with the primary endpoint being ORR. Preliminary results showed an ORR of 17.4%, with 34.8% of patients exhibiting stable disease. The median period of response was 13.5 months [66]. Similar to various solid tumor types, more studies would be needed in order to elucidate the prognostic impact of PD-L1 expression as a biomarker of response to ICIs in patients with PCa.

In general, PCa has a relatively low expression level of PD-L1 [73]. Bishop et al. investigated whether the targets of immunotherapy, namely PD-1, PD-L1/2, and CTLA-4, are overexpressed in patients with mCRPC who were resistant to enzalutamide. The results revealed that tumor resistance to enzalutamide is associated with an upregulation in the expression of PD-L1 [74]. Evidence of antitumor activity was documented in a phase II study in which the anti-PD-1 antibody pembrolizumab was added to patients receiving enzalutamide treatment who had progressive disease [75]. Three (30%) patients experienced a rapid decline in serum PSA level to below 0.2 ng/mL. Two patients with measurable disease at enrolment both showed a partial response. One of the partial responders was revealed to be MSI-positive, which validates the observations of previous reports of MSI as a prognosticator of response to anti-PD-1 antibodies [76]. Although only one patient in the aforementioned trial of pembrolizumab had PCa, this ICI is now approved for the treatment of patients with MSI-positive tumors. Table 1 summarizes clinical trials evaluating the clinical efficacies of ICI monotherapies.

### 4.2. Combination Immunotherapy Regimens

Several studies have investigated the efficacy of anti-PD-1 inhibitors in combination with various pharmacotherapeutic regimens for the treatment of advanced PCa. The two most promising combination immunotherapeutics for the treatment of mCRPC are the combined usage of two individual ICIs or one ICI with the augmentation of enzalutamide, a novel androgen receptor-axis-targeted agent approved for mCRPC. The phase II CHECKMATE-650 (NCT02985957) clinical trial administered a combination of the PD-1 inhibitor nivolumab with the anti-CTLA4 antibody ipilimumab as a second- or third-line therapy for patients with asymptomatic or minimally symptomatic mCRPC [77]. In this clinical trial, the study population was stratified into two arms based on the previous administration of taxane-based chemotherapy. A total of 78 patients were enrolled, with a minimum observational period of six months. A 10% ORR was observed in the combined ICI arm, which included patients who previously received chemotherapy. However, a superior ORR of 26% was observed in patients who were chemotherapy-naïve [77].

In the setting of PCa in which ICI monotherapy has shown limited success, the combination of nivolumab and ipilimumab showed promising clinical results. Further investigations regarding this combination would likely explore alternative dosages and administration intervals of both ICIs. Ongoing studies are determining whether this combination would be effective for a different subgroup of patients. In 2012, an open-labeled phase I dose-escalation trial (NCT01510288) evaluated ipilimumab in combination with the granulocyte-macrophage colony-stimulating factor-transduced cell-based allogeneic prostate cancer vaccine (GVAX) [78]. Patients with chemotherapy-naïve mCRPC enrolled in the study showed promising antitumor activity, with 25% of the patients exhibiting a PSA decline of greater than 50% and overall good tolerance to the agent [78]. The prospective phase II STARVE-PC trial (NCT02601014) enrolled patients with metastatic PCa with androgen receptor splice variant 7 (AR-V7) positive phenotypes [79]. Patients were administered nivolumab 3 mg/kg with ipilimumab 1 mg/kg every 3 weeks for four doses, and maintenance therapy was performed with nivolumab 3 mg/kg every 2 weeks. For the determination of DNA repair deficiency, targeted next-generation sequencing was utilized. The outcomes were generally more favorable in the patient population with AR-V7-positive PCa with DNA repair deficiency. The currently ongoing single-arm phase II NEPTUNES clinical trial (NCT03061539) is investigating the efficacy of ICI combination therapy in patients with higher tumor mutation burden (TMB) due to a defective DNA damage response (dDDR) or DNA mismatch repair gene mutations (dMMR) and in patients with high tumor-infiltrating lymphocytes [80].

Improvements in our molecular biology knowledge have shown that the AR signaling pathway is a critical element in CRPC progression and is, therefore, a rational target for the treatment of advanced disease and CRPC. Novel combination regimens involving PD-1 inhibitors with enzalutamide have shown promising results in this clinical field [81]. KEYNOTE-365 (NCT02861573) was a phase Ib/II umbrella study that tested various combination therapies in patients with CRPC [82]. The study cohort consisted of 69 patients with mCRPC who have progressed on chemo-naïve abiraterone and were treated with pembrolizumab and enzalutamide. Primary endpoints were safety and PSA response rate. The study results revealed an ORR of 20% and a PSA response rate of 33% [82]. A phase II trial (NCT02312557) enrolled 58 patients with chemotherapy-naïve mCRPC whose disease had progressed during enzalutamide and received four doses of 200mg IV pembrolizumab every three weeks while still on enzalutamide [75]. The primary endpoint was a PSA response greater than 50% (PSA_50_). Five of 28 (18%) patients achieved PSA_50_, while three of 12 patients (25%) with measurable disease at baseline had objective responses documented by radiographic imaging [75]. With promising results in the aforementioned clinical trials, KEYNOTE-641 (NCT03834493), a randomized double-blind phase III trial, has started accrual of patients with mCRPC as of July 2019 to evaluate the efficacy and therapeutic safety of the pembrolizumab and enzalutamide combination vs. enzalutamide plus placebo [83]. Table 2 summarizes clinical trials evaluating the clinical efficacies of combination immunotherapies.

### 4.3. The Utilization of Genomic Selection for Checkpoint Inhibitor Immunotherapy

Another strategy for the use of ICIs in the treatment of PCa focuses on specifically tailored patient selection. This approach of patient selection first started with colon cancer patients, as these patients also exhibit limited treatment response to ICI monotherapy [76]. Investigators noted that patients with dMMR variants of colon cancer have more somatic mutations compared to those with MMR-proficient tumors and higher immune infiltration [76]. A breakthrough study demonstrated the clinical importance of MMR gene mutations and analyzed clinical response to pembrolizumab monotherapy in cancers with and without dMMR variants. Patients with dMMR cancers exhibited an ORR of 40% and a 12-month PFS rate of 78%, while those with MMR-proficient colorectal cancer showed no objective responses and a 12-month PFS rate of 11% [84]. Based on this finding, pembrolizumab was FDA-approved in 2017 for the treatment of unresectable or metastatic solid organ malignancies of any histologic origin exhibiting dMMR or MSI-high expression [85]. This FDA-approval of pembrolizumab was the first cancer treatment to be approved based on a biomarker instead of the primary site of origin [85].

According to a clinical genomic analysis reported by Robinson et al., 3% to 5% of advanced PCa expresses MMR gene mutations, which supports the notion that this disease spectrum could be susceptible to pembrolizumab [11]. To date, retrospective studies have analyzed the therapeutic efficacy of PD-1 inhibitors in patients with mCRPC with dMMR mutations. Antonarakis et al. utilized the Johns Hopkins somatic genomic database for pathogenic loss of function MMR mutations in 13 patients with a deleterious mutation in the MMR gene. Results showed that six (46%) patients had *MSH6* mutations, while two of four (50%) patients who had received PD-1 inhibitors showed a clinically meaningful PSA_50_ [70]. Interestingly, this subset of patients was also observed to be sensitive to standard ADT, exhibiting a ≥90% PSA response rate of 85% and a median PSA PFS of 55 months [70]. In another study, 1551 tumors from 1346 patients with PCa were reviewed. Among 1033 patients who were eligible for the study, 32 (3.1%) patients with mCRPC exhibited MSI-high or dMMR tumors. Among these patients, 11 (34.4%) had been treated with a single-agent anti-PD-1 or PD-L1 treatment. It was observed that six (54.5%) patients achieved PSA_50_, and four (36.4%) patients showed a radiographic response [69]. Although the MSI-high/dMMR phenotype is uncommon and the subset of this PCa population is small, the biomarker-specific approval of pembrolizumab and the clinical evidence of therapeutic efficacy for this specific subgroup of patients with PCa supports the prospect of tumor sequencing for MMR deficiency in all patients with mCRPC, as recommended in the 2019 version of the National Comprehensive Cancer Network Guidelines [86].

Along with the MSI-high/dMMR genetic expression, clinical trials are ongoing to discover other molecular subtypes that may respond to ICIs. In the open-label phase II KEYNOTE-199 (NCT02787005) trial, the clinical efficacy of pembrolizumab for patients with chemotherapy-resistant mCRPC was assessed in three parallel cohorts [64]. Pembrolizumab monotherapy showed a 12% ORR in patients with somatic *ATM* or *BRCA1/2* mutations, which was higher than the ORR of 4% for patients without the aforementioned mutations [64]. A comparable trend was observed in the CHECKMATE-650 trial, in which a superior response rate was observed in patients with homologous repair-deficient mCRPC [77]. The ImmunoProst (NCT03040791) study is a multicenter, single-arm, open-label phase II trial of nivolumab in patients with mCRPC who have progressed to a previous taxane-based chemotherapy regimen. The study will analyze germline and somatic DNA repair defects in that patient cohort [87]. Another multicenter phase II trial (NCT03248570) is investigating the effects of pembrolizumab in patients with mCRPC stratified according to the presence of defects in DNA damage repair [88].

There is accumulating clinical evidence that a subgroup of patients with CDK12 inactivation may benefit from immunotherapy. Due to numerous focal tandem duplications and increased gene fusion activity, along with an upregulation of neoantigens and T cell infiltration, solid malignancies with CDK12 biallelic mutations have been suggested to harbor a distinctive immune signature [89]. This observation infers the existence of another potential genomic subtype of PCa that could be susceptible to ICIs [90]. Various studies estimate that 4.7% of patients with mCRPC harbor this mutation, making PCa the tumor population with the greatest distribution of biallelic inactivation of CDK12 among all solid tumors [89,91,92]. Wu et al. reported the efficacy of PD-1 inhibitor treatment in four patients with mCRPC harboring CDK12 mutations and showed that two (50%) patients showed antitumor activity in terms of PSA decline [89]. A recently conducted multicenter trial in patients with advanced PCa with loss of function CDK12 mutations evaluated the efficacies of the poly (ADP-ribose) polymerase and PD-1 inhibitors. Three of eight (38%) patients with CDK12-mutated mCRPC had a significant PSA decline, with a median PFS of 6.6 months [93]. The ongoing IMPACT (NCT03570619) trial is investigating whether this subtype of mCRPC is more susceptible to a combination of ICIs consisting of ipilimumab and nivolumab followed by nivolumab monotherapy [94].

With advances in next-generation tumor DNA sequencing, new subsets of patients with PCa are being identified through precision medicine. POLE mutations, or mutations in the exonuclease domain of the proofreading DNA polymerase epsilon enzyme, have been observed in patients with PCa. These mutation phenotypes were observed despite an MSI-stable signature [95]. Although POLE mutations are reported to affect only 0.1% of patients with mCRPC, this gene mutation may have potential therapeutic efficacy with anti-PD1 antibodies due to the vast quantity of predicted mutation-associated neoantigens [96].

Despite recent advances and achievements of checkpoint inhibition and other modalities of immunotherapy, inconsistencies continue to occur between treatment and response across the different tumor spectra. PCa develops complex genotype mutations that significantly affect the outcome of immunotherapy [97]. The classification and identification of patients who will successfully respond to immunotherapy is of paramount importance and is dependent on the future development of biomarkers. It is crucial to establish reliable biomarkers that can aid the initial decision to determine which individual group of patients will be more responsive to a specific mode of immunotherapy. Clinical trials focusing on genomic and DNA transcription data are ongoing to unravel immune signatures to aid the prediction of therapeutic efficacy and response [97,98,99].

### 4.4. Adoptive Cellular Therapy in the Treatment of Prostate Cancer

Adoptive cellular therapy (ACT) is an emerging treatment strategy in immunotherapy, in which the autologous T cells of the patient are extracted and genetically manipulated prior to reinfusion. Various approaches for ACT, such as engineered T cell receptors, tumor-infiltrating lymphocytes, and chimeric antigen receptors (CAR) T cells, have been developed for the treatment of PCa patients with metastases or in advanced disease settings [100]. Of the aforementioned engineered T cells, CAR-T cells have shown the most therapeutic potential. These cells are genetically constructed to express a specific CAR gene ex vivo to target antigens utilizing an antibody derived single-chain variable fragment (scFv) [101]. After infusion, CAR-T cells activate an inflammatory reaction that results in the cytolysis of the antigen-expressing tumor cell [102].

In patients with PCa, CAR-T cells are used to target prostate-specific membrane antigen (PSMA), which is a type II transmembrane glycoprotein that is specifically upregulated in PCa cells [103]. In a phase I clinical trial conducted by Junghans et al., “designer” CAR-T cells targeting PSMA were continuously infused with a low dose of IL-2 [104]. Three of the five enrolled patients met the 20% engraftment target of CAR-T cells, and there were no anti PSMA toxicities, while two of the three engraftment successful patients exhibited a PSA50 response [104]. In another phase I trial, the safety and dosage of modified autologous PSMA targeted T cells was analyzed. The study concluded that a dosage up of 3 × 107 cells/kg could be safely administered, with persistence of T cell activity lasting for up to 2 weeks [105]. Interestingly, one patient experienced a long-term response over 16 months, which suggests clinical activity [105].

Similar to other cellular-based immunotherapies, CAR-T cell therapy elicits limited therapeutic responses in the treatment of solid tumors, which includes PCa [101]. The major obstacle in CAR-T cell therapy is the tumor microenvironment, which has immunosuppressive properties. To overcome this hindrance, a PSMA directed/TGF-β insensitive CAR-T cell is undergoing a phase I clinical trial (NCT03089203) [106]. The distinctive TGF-β insensitive features of this cell product have the potential to show therapeutic efficacy in the immunosuppressive PCa microenvironment caused by high levels of TGF-β [106]. With the occurrence of neoantigens with resistant clones, further clinical studies and research focusing on next-generation CAR-T cells with multiple scFv targeting multiple tumor-specific antigens are warranted.

## 5. Conclusions

Considerable progress has been made in the immunotherapeutic approach to the treatment of PCa. However, further research is still warranted to fully understand the immunological mechanisms involved in the PCa microenvironment. For now, the treatment benefits of ICIs show promise and potential. Along with standard monotherapeutic strategies utilizing novel ICIs, the efficacy of the combined usage of two individual ICIs and the administration of one ICI with standard therapies still needs to be elucidated. In the present clinical setting, these regimens will likely be used in some capacity and show therapeutic benefit in a select group of patients with PCa. In the near future, there will be other classes of immunotherapeutic agents, which may become an appropriate treatment for patients with mCRPC. For now, the classification of patients and administration modalities of novel immunotherapeutic agents remain unsolved. Moreover, studies are warranted to investigate how the modification in the sequencing of prior agents could affect the efficacy of the novel agents.

## Figures and Tables

**Figure 1 ijms-21-05484-f001:**
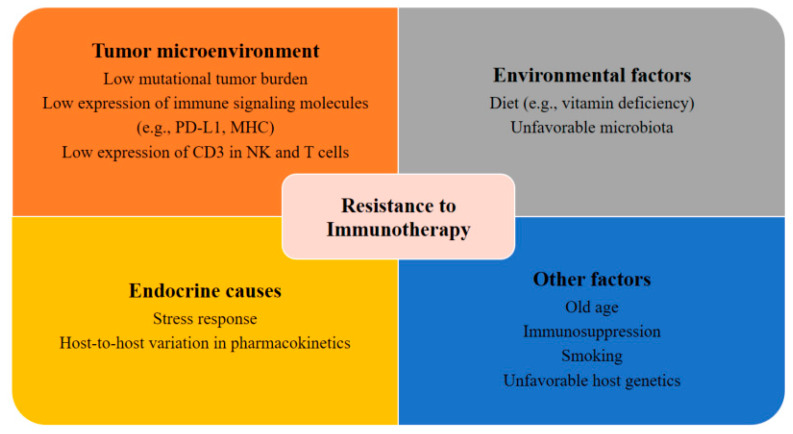
Factors contributing to resistance to immunotherapy. Several potential tumor-related, host-related, and environmental factors may elicit resistance to immunotherapies.

**Figure 2 ijms-21-05484-f002:**
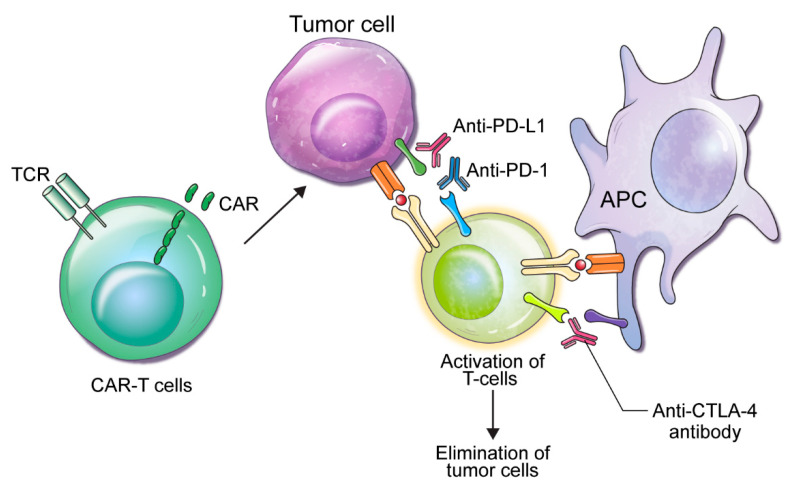
Molecular mechanisms of CTLA4 and PD-1 attenuation of T cell activation. Schematic diagram of the molecular interactions and downstream signaling induced by ligation of CTLA4 and PD-1 by their respective ligands. APC: antigen-presenting cell, CAR: chimeric antigen receptor, CTLA-4: cytotoxic T lymphocyte-associated protein 4, PD-1: programmed death 1, PL-L1: programed death ligand 1, TCR: T cell receptor.

**Table 1 ijms-21-05484-t001:** Clinical trials evaluating immune checkpoint inhibitor monotherapy for prostate cancer.

Agent	Mechanism	Clinical Phase	Identifier	Indication	Primary Endpoints
Ipilimumab	Immunotherapy + radiotherapy	I/II	NCT00323882 [72]	Metastatic CRPC	AE, PSA response, and tumor response
Ipilimumab	Immunotherapy	III	NCT00861614 (CA184-043) [67]	Metastatic CRPC, post-docetaxel	OS
Ipilimumab	Immunotherapy	III	NCT01057810 [65]	Metastatic CPRC, chemotherapy-naïve	OS
Nivolumab	Immunotherapy	Ib	NCT00730639 (MDX-1106) [68]	CRPC	Safety, antitumor activity, and pharmacokinetic properties
Pembrolizumab	Immunotherapy	Ib	NCT02054806 (KEYNOTE-028) [66]	Advanced prostate cancer with PD-L1 expression ≥ 1% of tumor or stromal cells	ORR

AE: adverse event; CRPC: castration-resistant prostate cancer; ORR: objective response rate; OS: overall survival; PD-L1: programmed death-ligand 1; PSA: prostate-specific antigen.

**Table 2 ijms-21-05484-t002:** Clinical trials evaluating combination immunotherapies for prostate cancer.

Combination Agents	Mechanism	Clinical Phase	Trial ID	Indication	Primary Endpoints
Ipilimumab + nivolumab	Dual checkpoint blockade	II	NCT02985957 (CHECKMATE-650) [77]	Metastatic CRPC	ORR and rPFS
Ipilimumab + GVAX	Vaccination + immunotherapy	I	NCT01510288 [78]	Metastatic CRPC	AE
Ipilimumab + nivolumab	Dual checkpoint blockade	II	NCT02601014 (STARVE-PC) [79]	Metastatic CRPC with detectable AR-V7 transcript	PSA response
Ipilimumab + nivolumab	Dual checkpoint blockade	II	NCT03061539 (NEPTUNES) [80]	Metastatic CRPC with TMB	CRR
Pembrolizumab + enzalutamide	Checkpoint blockade + ADT	Ib/II	NCT02861573 (KEYNOTE-365) [82]	Metastatic CRPC	AE, PSA response, ORR
Pembrolizumab + enzalutamide	Checkpoint blockade + ADT	III	NCT03834493 (KEYNOTE-641) [83]	Metastatic CRPC	OS and rPFS

ADT: androgen-deprivation therapy; AE: adverse event; CRPC: castration-resistant prostate cancer; CRR: composite response rate; ORR: objective response rate; OS: overall survival; PSA: prostate-specific antigen; rPFS: radiographic progression-free survival; TMB: tumor mutation burden.

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
