# Peer review of "Current Status and Future Perspectives of Checkpoint Inhibitor Immunotherapy for Prostate Cancer: A Comprehensive Review"

_ijms, 2020, doi:10.3390/ijms21155484_

Round 1
Reviewer 1 Report
Authors present a quality and well-written review that describes current status and future perspectives of checkpoint inhibitor immunotherapy for prostate cancer. Authors highlight the re-emergence of immunotherapeutic approaches in the treatment of PCa, mainly focusing on immune checkpoint inhibitors (ICIs), which have shown the potential to revolutionize the treatment of both localized and advanced PCa.
Authors discuss recent progress pathophysiology of prostate cancer, immunotherapy resistance of prostate cancer, immune checkpoint inhibitors for the treatment of prostate cancer (including ICI monotherapy, combination immunotherapy regimens, the utilization of genomic selection for checkpoint inhibitor immunotherapy,
Comments:
- Table 1. Please correct the column widths not to have single letters on separate lines.
- The authors are strongly recommended to add 1-2 figures.
e.g. Fig 1 – Checkpoint immunotherapy in general (overview of various regimens); Fig 2 – mechanisms of resistance development.
- Authors might consider briefly mentioning other immunotherapy approaches, such as CAR-T and TCR-T.
- Authors are kindly requested to cite the following article (DOI 10.3390/cancers12010125) that describes immunotherapy approaches for the treatment of solid tumors.
Overall, the manuscript is valuable for the immunotherapy scientific community and should be accepted for publication, after minor edits are made and figures are added.
Author Response
Authors present a quality and well-written review that describes current status and future perspectives of checkpoint inhibitor immunotherapy for prostate cancer. Authors highlight the re-emergence of immunotherapeutic approaches in the treatment of PCa, mainly focusing on immune checkpoint inhibitors (ICIs), which have shown the potential to revolutionize the treatment of both localized and advanced PCa. Authors discuss recent progress pathophysiology of prostate cancer, immunotherapy resistance of prostate cancer, immune checkpoint inhibitors for the treatment of prostate cancer (including ICI monotherapy, combination immunotherapy regimens, the utilization of genomic selection for checkpoint inhibitor immunotherapy. Overall, the manuscript is valuable for the immunotherapy scientific community and should be accepted for publication, after minor edits are made and figures are added.
REPLY: We are very much thankful to the reviewer for the thorough review. We agree with all the specific comments raised and have revised our paper in light of the useful suggestions. Our responses to the specific comments/suggestions/queries given are provided below.
Table 1. Please correct the column widths not to have single letters on separate lines.
REPLY: We agree with the reviewer that the column widths are not uniform. According to the reviewer’s recommendation, we have adjusted the columns for reading clarity.
The authors are strongly recommended to add 1-2 figures. e.g. Fig 1 – Checkpoint immunotherapy in general (overview of various regimens); Fig 2 – mechanisms of resistance development.
REPLY: Thank you for this valuable comment. We have added a figure (Fig. 1) illustrating the general mechanisms of checkpoint inhibition and other immunotherapeutic modalities. In addition, figure 2 shows the resistance mechanism of PCa cells to immunotherapy. We believe that our diagrams will aid the reader in understanding the pathophysiologic mechanisms of immunotherapy in PCa.
Authors might consider briefly mentioning other immunotherapy approaches, such as CAR-T and TCR-T.
REPLY: Recently, adoptive cellular therapy (ACT), an approach in which a patient’s T cells are removed and genetically modified ex vivo before reinfusion, has gained attention in the field of immunotherapy. Several variations of ACT have been in development, including tumor-infiltrating lymphocytes (TILs), engineered T cell receptors (TCR), and chimeric antigen receptors (CAR) T cell therapy. Recent studies have shown promising results for CAR-T cell therapy in solid tumors, including PCa. For the treatment of PCa, CAR-T cells are targeted at prostate-specific membrane antigen (PSMA). Such treatments show clinical promise and may hold potential for the treatment of PCa in advanced clinical settings. Collectively, these advances may open new horizons in the ongoing battle towards developing treatment approaches for the disease continuum of PCa. In line with the reviewer’s suggestions, we have added another section in our paper under the heading of 4.4 adoptive cellular therapy in the treatment of prostate cancer. Thank you for your valuable insight.
Authors are kindly requested to cite the following article (DOI 10.3390/cancers12010125) that describes immunotherapy approaches for the treatment of solid tumors.
REPLY: The aforementioned article was cited under 4.4 adoptive cellular therapy in the treatment of prostate cancer in our Discussion section. Thank you for your suggestion.
Reviewer 2 Report
General:
This is a comprehensive review of the present status of immune checkpoint inhibitor (ICI) immunotherapy of prostate cancer (PCa), applied either as monotherapy or as combined therapy.
The authors first give a brief overview of current agents and protocols approved by the FDA for the treatment of PCa. They continue with a description of the pathophysiology of PCa, putting special emphasis on the cellular composition of the tumor microenvironment, the presence of immune mediators and the potential role of inflammation in disease initiation and progression. The authors then refer to the basis for resistance of PCa to immunotherapy and move to a detailed survey of current studies employing ICI for the treatment of PCa. The authors discuss the correlation of clinical response (or rather lack thereof) with the mutational load in PCa and further elaborate on the fraction of tumors exhibiting an MSI signature. The prospects of combination therapy with two ICI reagents (anti-CTLA-4 and anti-PD-1/PD-L1) or of an ICI and reagents affecting the androgen receptor axis (emphasis on enzalutamide) are also discussed in depth.
Overall this is an excellent, updated and well-written review of this topic, which is of wide interest to the readers of IJMS.
I highly recommend that the author add a paragraph referring to the prospects of combining ICI with cellular immunotherapies such as CAR-T cell therapy for PCa.
Minor comments:
- Page 3, 2nd paragraph: The sentence: “Cancer cells exhibit tumor self-antigens that are immunogenic, in which most do not express foreign antigens” is not clear.
- Page 3: Please clarify the relevance of: “This view is still under debate, with a recent genomic analysis showing that even patients with renal cancer carcinoma (RCC) had a high mutational burden.”
- Please describe ‘enzalutamide’ in the first time it appears in the text rather than in page 5.
- Bottom of page 5: Please explain to the reader what AR-V7 stands for.
- Concerning mg/kg specifications, please make sure to adhere to the format ‘3 mg/kg’ (i.e., correct slash and space). See, for example, the top two lines in page 6.
- Please adhere to CDK12 (rather than CDK 12).
Author Response
This is a comprehensive review of the present status of immune checkpoint inhibitor (ICI) immunotherapy of prostate cancer (PCa), applied either as monotherapy or as combined therapy. The authors first give a brief overview of current agents and protocols approved by the FDA for the treatment of PCa. They continue with a description of the pathophysiology of PCa, putting special emphasis on the cellular composition of the tumor microenvironment, the presence of immune mediators and the potential role of inflammation in disease initiation and progression. The authors then refer to the basis for resistance of PCa to immunotherapy and move to a detailed survey of current studies employing ICI for the treatment of PCa. The authors discuss the correlation of clinical response (or rather lack thereof) with the mutational load in PCa and further elaborate on the fraction of tumors exhibiting an MSI signature. The prospects of combination therapy with two ICI reagents (anti-CTLA-4 and anti-PD-1/PD-L1) or of an ICI and reagents affecting the androgen receptor axis (emphasis on enzalutamide) are also discussed in depth. Overall this is an excellent, updated and well-written review of this topic, which is of wide interest to the readers of IJMS.
REPLY: We are very much thankful to the reviewer for the thorough review. We agree with all the specific comments raised and have revised our paper in light of the useful suggestions. Our responses to the specific comments/suggestions/queries given are provided below.
I highly recommend that the author add a paragraph referring to the prospects of combining ICI with cellular immunotherapies such as CAR-T cell therapy for PCa.
REPLY: Recently, adoptive cellular therapy (ACT), an approach in which a patient’s T cells are removed and genetically modified ex vivo before reinfusion, has gained attention in the field of immunotherapy. Several variations of ACT have been in development, including tumor-infiltrating lymphocytes (TILs), engineered T cell receptors (TCR), and chimeric antigen receptors (CAR) T cell therapy. Recent studies have shown promising results for CAR-T cell therapy in solid tumors, including PCa. For the treatment of PCa, CAR-T cells are targeted at prostate-specific membrane antigen (PSMA). Such treatments show clinical promise and may hold potential for the treatment of PCa in advanced clinical settings. Collectively, these advances may open new horizons in the ongoing battle towards developing treatment approaches for the disease continuum of PCa. In line with the reviewer’s suggestions, we have added another section in our paper under the heading of 4.4 adoptive cellular therapy in the treatment of prostate cancer. Thank you for your valuable insight.
Page 3, 2nd paragraph: The sentence: “Cancer cells exhibit tumor self-antigens that are immunogenic, in which most do not express foreign antigens” is not clear.
REPLY: We agree with the reviewer that this sentence needs clarification. Tumor immunogenicity is a major determinant of antigen-specific antitumor activity. Unlike pathogens that express foreign antigens, cancer cells belong to the self-immune system and mostly do not express foreign antigens. On the other hand, they do exhibit tumor self-antigens that are potentially immunogenic. On par with the reviewer’s suggestion, we have edited the sentence for better understanding. Thank you for this comment.
Page 3: Please clarify the relevance of: “This view is still under debate, with a recent genomic analysis showing that even patients with renal cancer carcinoma (RCC) had a high mutational burden.”
REPLY: Tumor mutation burden, and presumably neoantigen load, has been shown to be correlated with immunotherapeutic responses; however, this is not always the case as tumor immunogenicity may also be dependent on several other factors that are modulated by cancer cells or cells within the tumor microenvironment. PCa is generally considered to be poorly immunogenic and to present with a lower mutation burden, compared with cancers associated with mutagenic exposure, such as skin, melanoma, lung, and bladder cancers. However, as we have stated, this view is controversial, as stratified genomic analysis has indicated that PCa patients have a higher mutational burden than that in renal cancer patients. We have edited our manuscript for literary fluidity and clarity. Thank you for your valuable comment.
Please describe ‘enzalutamide’ in the first time it appears in the text rather than in page 5.
REPLY: Thank you for your insight. To accommodate the reviewer’s recommendation, we have edited the introduction section of our manuscript as follows: Current agents approved by the US Food and Drug Administration (FDA) that have shown efficacy in terms of overall survival (OS) in patients with mCRPC include docetaxel, cabazitaxel, radium-223, sipuleucel-T, and androgen receptor axis-targeted agents including abiraterone and enzalutamide.
Bottom of page 5: Please explain to the reader what AR-V7 stands for.
REPLY: We agree with the reviewer that the full term of AR-V7 needs to be stated. We have added the full terminology of androgen receptor splice variant 7 for clarification. Thank you for this comment.
Concerning mg/kg specifications, please make sure to adhere to the format ‘3 mg/kg’ (i.e., correct slash and space). See, for example, the top two lines in page 6.
REPLY: Thank you for your insight. We have edited the manuscript accordingly.
Please adhere to CDK12 (rather than CDK 12).
REPLY: As the reviewer has suggested, we have edited our manuscript. Thank you for the comment.
Reviewer 3 Report
Kim and Koo provide a comprehensive overview of the checkpoint immunotherapy for prostate cancer. The review is well written. My only suggestion is that the authors should include an overview figure to illustrate the agents and their targets and draw the relevant pathways with the tumor as the background.
Author Response
Kim and Koo provide a comprehensive overview of the checkpoint immunotherapy for prostate cancer. The review is well written. My only suggestion is that the authors should include an overview figure to illustrate the agents and their targets and draw the relevant pathways with the tumor as the background.
REPLY: We are very much thankful to the reviewer for the thorough review. We have added a figure (Fig. 1) illustrating the general mechanisms of checkpoint inhibition and other immunotherapeutic modalities. In addition, figure 2 shows the resistance mechanism of PCa cells to immunotherapy. We believe that our diagrams will aid the reader in understanding the pathophysiologic mechanisms of immunotherapy in PCa.